# Rapid Maxillary Expansion on the Adolescent Patient: Systematic Review and Case Report

**DOI:** 10.3390/children9071046

**Published:** 2022-07-14

**Authors:** Alessio Danilo Inchingolo, Irene Ferrara, Fabio Viapiano, Anna Netti, Merigrazia Campanelli, Silvio Buongiorno, Giulia Latini, Vincenzo Carpentiere, Anna Maria Ciocia, Sabino Ceci, Assunta Patano, Fabio Piras, Filippo Cardarelli, Damiano Nemore, Giuseppina Malcangi, Angela Di Noia, Antonio Mancini, Angelo Michele Inchingolo, Grazia Marinelli, Biagio Rapone, Ioana Roxana Bordea, Antonio Scarano, Felice Lorusso, Daniela Di Venere, Francesco Inchingolo, Gianna Dipalma

**Affiliations:** 1Department of Interdisciplinary Medicine, University of Bari “Aldo Moro”, 70124 Bari, Italy; ad.inchingolo@libero.it (A.D.I.); ire.ferra3@gmail.com (I.F.); viapianofabio96@gmail.com (F.V.); annanetti@inwind.it (A.N.); merigrazia.92@hotmail.it (M.C.); silvio.buongiorno@gmail.com (S.B.); dr.giulialatini@gmail.com (G.L.); vincenzo.carpentiere@gmail.com (V.C.); anna.ciocia1@gmail.com (A.M.C.); s.ceci@studenti.uniba.it (S.C.); assuntapatano@gmail.com (A.P.); dott.fabio.piras@gmail.com (F.P.); drfilippocardarelli@libero.it (F.C.); damianonemore@gmail.com (D.N.); giuseppinamalcangi@libero.it (G.M.); angedinoia@libero.it (A.D.N.); dr.antonio.mancini@gmail.com (A.M.); angeloinchingolo@gmail.com (A.M.I.); graziamarinelli@live.it (G.M.); daniela.divenere@uniba.it (D.D.V.); giannadipalma@tiscali.it (G.D.); 2Department of Oral Rehabilitation, Faculty of Dentistry, Iuliu Hațieganu University of Medicine and Pharmacy, 400012 Cluj-Napoca, Romania; 3Department of Innovative Technologies in Medicine and Dentistry, University of Chieti-Pescara, 66100 Chieti, Italy; ascarano@unich.it (A.S.); felice.lorusso@unich.it (F.L.)

**Keywords:** maxillary expansion, palatal expansion, adolescent, permanent dentition, adolescent patient

## Abstract

**Aim:** In the literature, many studies and articles are investigating new devices and approaches to achieve rapid palate expansion through the opening of the palatal suture, and evaluating the skeletal, dental, and soft tissue effects. The purpose of this review was to assess how palatal expansion is performed in adolescent patients with permanent dentition. Furthermore, it was reported as an example of successful orthodontic treatment of an 11-year-old female patient affected by maxillary skeletal transverse deficiency, in permanent dentition. **Methods:** A search of the literature was conducted on PubMed, Cochrane, Scopus, Embase, and Web of Science databases. Inclusion criteria were the year of publication between 2017 and 2022, patients aged 10 to 16 years in permanent dentition, with transversal discrepancy, treated with tooth-borne, bone-borne, hybrid palatal expanders. **Results:** A total of 619 articles were identified by the electronic search, and finally, a total of 16 papers were included in the qualitative analysis. **Conclusions:** From this study, it was assessed that MARPE is more predictable, and it determines a more significant expansion of the suture than the Hyrax expander, with fewer side effects.

## 1. Introduction

Rapid maxillary expansion (RME) is commonly used to treat transverse maxillary deficiencies, which are characterized by abnormally low maxillary growth. The etiological causes of this condition can be genetic or environmental, and it is frequently associated with dental crowding, crossbite, Class II and III malocclusion, and temporomandibular joint dysfunction [1,2].

Usually, RME is performed using the following: tooth-tissue-borne RME appliance (TTB RME), tooth-borne RME appliance (TB RME) (Figure 1), bone-borne RME appliance (BB MARPE) and tooth–bone-borne MARPE appliance (TBB MARPE) (Figure 2) [3].

Different protocols of activation of these appliances lead to the opening of the palatine suture and the increased width of the palate [4]. The median palatine suture extends from the anterior palatine fissure to the posterior nasal spine The ossification process of the suture is closely correlated with age and sex; it begins with the formation of spicules along the suture that increase during growth, creating firm interdigitations [5,6]. As the patient grow older, it becomes more difficult to open the suture [4]. Females presented greater density ratios of the midpalatal suture than males after expansion treatment [7].

CBCT is the most accurate tool for observing the dentoskeletal changes after RME.

Pasqua et al. stated that the skeletal changes are more significant with higher activations protocol. There is a correlation between the opening of the suture and the increase in nasal cavity shape [8].

In transverse discrepancies in adolescent patients, the most controversial decision is how to perform upper jaw expansion. Several papers have shown great variability in the timing of maturation and ossification of the midpalatal suture; for this reason, the choice between traditional rapid expander and skeletally anchored expander cannot be based only on the patient’s age [9]. Moreover, through cone-beam computed tomography (CBCT), it is possible to assess the bone density and the quality of interdigitation of the bony bridges of the palatine suture, an essential evaluation for the choice of appliance [10].

Before beginning expansion, the patient’s skeletal age needs to be well assessed, because very often it does not coincide with the age of birth. The amount of activation influenced the higher nasal skeletal changes in the Hybrid Hyrax group. [8]. Many studies show that younger patients could have a greater degree of skeletal maturation than what was expected based on their young age [11].

Schauseil et al. analyzed changes in midpalatal suture density after RME treatment using low-dose CT. Low-dose CT scans were taken at three different points of time: initially (T0), following maximum expansion (T1) and after six months of retention (T2). The sutural density was considerably lower following RME, while it significantly increased after six months of retention [12]. Contrary to this, Franchi et al. demonstrated that there was no statistically significant change in the sutural density at the start (T0) and after six months of retention in a younger group of patients (T2) [13].

Generally, two methods are used to assess the patient’s skeletal maturation. One is the degree of skeletal maturation of the cervical vertebrae belonging to Bacetti et al. The paper analyzed the validity of six stages of cervical vertebral maturation from Cvs1 through Cvs6, based on the shape and the concavity of the lower margin of the vertebrae, as a biological indicator for skeletal maturity to correlate vertebral stage with peak statural growth. The greatest increment in mandibular and craniofacial growth was during the interval from cervical vertebral stage 3 (Cvs3) to stage 4 (Cvs4). In the Cvs3 and Cvs4 stages, the bodies of all cervical vertebrae are rectangular, the inferior border of the third and fourth vertebra, respectively, develops a concavity, and the peak in statural height also occurred with a prevalence rate of 93.5% in the subjects examined [14].

Another method is the study of the palatine suture according to Angelieri et al.: the authors classified the palatine suture morphology and maturation by CBCT in 5 stages (A, B, C, D, and E). From stage C onward, the suture appears partially interdigitated; there are two parallel, jagged, high-density lines that are very close together, separated by small spaces and alternating with low bone density areas [10]. An initial diagnosis of stage C indicates an uncertain prognosis for performing a traditional rapid expansion, as the onset of fusion of the palatine portion of the suture may be imminent [15].

Some procedures for the correction of transverse deficits in adolescent orthodontic patients include a virtual planning phase for miniscrew-assisted rapid palatal expansion (MARPE) [16]. In detail, the insertion of orthodontic mini-implants (TADs) appears to be easier, more precise, and safer using digital systematics [17,18,19,20,21].

The aim of this systematic review is to evaluate the effectiveness and efficiency of fixed palatal expansion devices in adolescent patients with superior transverse deficit by comparing MARPE and traditional ERP.

## 2. Materials and Methods

### 2.1. Protocol and Registration

The Preferred Reporting Items for Systematic Reviews and Meta-Analyses (PRISMA) guidelines were used in this systematic review [22]. The review protocol was registered at PROSPERO under the unique number CRD42022334782.

### 2.2. Eligibility Criteria

In the research, the following studies were considered eligible: on adolescents aged 10 to 16 who had permanent dentition and transverse maxillary deficit, treated with rapid maxillary expansion (RME), which included all types of tooth-borne (TB) and tooth-tissue-borne (TTB) RME appliance, and MARPE, which included all types of MARPE appliance designs, whether hybrid tooth–bone-borne (TBB) or only bone-borne (BB) and expansion procedure. The following outcomes were considered eligible: the success rate of transverse maxillary expansion procedures (dental or skeletal), or any of the additional outcomes: duration, side effects (dental or periodontal), or soft tissue effects. Both randomized and non-randomized clinical trials and observational studies, either prospective or retrospective, were considered eligible.

### 2.3. Inclusion and Exclusion Criteria

Inclusion criteria were: randomized clinical trials (RCT), retrospective and observational studies; adolescent patients aged 10 to 16 with a transverse maxillary deficit; treatment performed with MARPE (bone-borne or hybrid appliances); compared to conventional RME (tooth-borne appliances). The outcomes analyzed markers of transverse expansion effectiveness and undesirable effects.

Studies that fulfill at least one of the following exclusion criteria were excluded: reviews, case series, letters, or comments; animal models or dry skulls studies; papers with no comparative data; or patients with previous or continuing orthodontic treatment, craniofacial syndromes, or cleft lip and palate.

### 2.4. Data Sources and Search Strategy

The qualifying criteria were developed using the PICOS (Population, Intervention, Comparison, Outcomes, and Study Design) framework. From 28 April 2017 to 28 April 2022, a systematic search was conducted in the PubMed, Cochrane, Scopus, Embase, and Web of Science databases. Keywords used were “maxillary expansion” or “palatal expansion” and “adolescent” or “permanent dentition”. Papers in the English language were selected. Table 1 summarizes the search approach in detail. The authors checked the titles and complete texts of any papers that might be relevant.

PICO question was: adolescent patients aged 10 to 16 with a transverse maxillary deficit (Population), orthodontic maxillary expansion treatment performed with MARPE, bone-borne or hybrid appliances (Intervention), compared to conventional RME (Comparison) and transverse expansion and undesirable effects after treatment (Outcome).

### 2.5. Data Collection

The study data was selected by analyzing type of study, age, gender, appliance, activation technique, maxilla and tooth width variations, and the skeletal and dental outcomes of the studies included at the maxillary molar level. 

## 3. Results

### Study Selection and Characteristics

The selection process is summarized in Figure 3. The electronic database search identified a total of 619 (Scopus *N* = 279, PubMed *N* = 214, Web of Science *N* = 53, Cochrane Library *N* = 47, Embas *N* = 26) and no articles were included through the hand search. After duplicate removal, 312 studies underwent title and abstract screening. In total, 272 papers were not selected after the abstract screening, mostly because of the inclusion of patients under the age of 10 or over the age of 16. Thirty-four articles were chosen for the eligibility assessment. Subsequently, 18 papers were eliminated after the full-text evaluation because they did not meet the inclusion criteria: 11 were off-topic, 4 were on younger patients, and 3 were case reports. Finally, 16 articles were picked for the systematic review (Figure 3).

The study design of the chosen studies was: eight randomized controlled clinical trials, seven retrospective studies, and one observational study. All the studies selected analyzed the consequences of the RME therapy: most of them were on teeth and skeletal tissues, and some of them evaluated changes in soft tissues and upper airways. The study’s sample size ranged from 20 to 60 people with an average age range from 11 to 16 years. Different appliances were used: TB, TTB, TBB MARPE, and BB MARPE.

The number of activations suggested was almost the same in all designs: in two studies, the screw was turned two times a day, for the first week, then the appliance was activated one time per day; the screw used was a 9 mm Hyrax for the TB group and a 9 mm Jackscrew for the MARPE TTB group [23,24]. In 11 studies suggested activating the screw two times a day until the finish of the expansion: the screws used in these cases were 9 mm Hyrax in the 9 TB group; a Hyrax miniscrew was used in the 1 TB group and 12 mm Hyrax jackscrew in the 1 TB group; one group used the Keles Keyless screw; in the MARPE TTB groups, two studies used 8 mm a miniexpander jackscrew and one study used a 12 mm jackscrew [25,26,27,28,29,30,31,32,33,34,35]. Two studies suggested one activation/day with the Hyrax screw [36,37], and the protocol was not determined in one study [38]. The activation cycle was completed when contact between the mesiopalatal cusps of the upper molars and the buccal cusps of the lower molars was reached. The appliance was subsequently kept as a retainer from 3 to 12 months: 3 months in eight studies, 6 months in seven studies and 12 months in one study. Data analyzed at T0 (pre-treatment) and T1 (post-treatment) were: CBCT images in nine studies; 3D dental models in three studies; micro-CT of teeth in two studies; cephalometric radiographs in one study and intraoral scans of palatal rugae in one study.

## 4. Case Report

### 4.1. Etiology and Diagnosis

This is the case report of an 11-year-old female patient affected by maxillary skeletal transverse deficiency, and mild mandibular and maxillary crowding. Her chief complaint was the impossibility of closing properly the mouth and the anesthetic appearance of her smile. The patient referred to menarche that occurred six months before therapy began.

Facial analysis showed an oval face, longer third face height, dental midline centered, and circumoral muscle strain on lip closure. The profile was convex with retruded mandible, poor definition of the chin and skeletal class II with hyperdivergent growth pattern. At the functional examination, it did not result in any signs of temporomandibular dysfunction, and no abnormalities in tongue volume or position were found (Figure 4).

Dental analysis showed permanent dentition in a molar Class I molar and canine on both sides. The centric relation mounting of the patient’s mouth showed a slightly post-rotation on the mandible due to precontact of the 1st quadrant, revealing a Class II molar and canine on the right side, and a Class I molar on the left. Overbite (2 mm) and overjet (4.7 mm) were mildly increased, and dental midlines were centered. In the occlusal view, it is possible to have an ogival-shaped palate, with the presence of dental compensation (vestibular tipping) on the maxillary first molars (UR6 and UL6), and both canines erupted ectopically slightly mesial. The occlusal view of the lower arch shows constricted mandibular arch, with the presence of dental compensation (lingual tipping) on the first mandibular molars (LR6 and LL6). Both arches had reduced transversal dimensions (Figure 1). Both arches presented crowding (4.5 mm in the maxillary and 3 mm in the mandibular arch), with a reduced trans-palatal width. Both curves of Spee and Wilson were slightly accentuated (2 mm and 1.5 mm deep) (Figure 5).

Lateral cephalometric tracing showed a skeletal Class II (ANB, 6.2°) and a hyperdivergent skeletal pattern (S-Go/N-Me, 57.5%). The inclination of the maxillary incisors was correct (U1- PP, 110.4°); instead, the mandibular incisors were slightly proclined (IMPA, 96.1°; Figure 5; Table 2).

The panoramic radiograph revealed a complete dentition, including all third molars that had not yet erupted, as well as no skeletal anomalies of the temporomandibular joint (Figure 6). 

Clinical evaluation revealed pathology during joint function.

### 4.2. Treatment Objectives

The primary goal of this treatment was the correction of transverse skeletal maxillary discrepancy. Even though the crossbites were not clinically visible, the reduced trans-palatal width (43, 42 mm) with a deep mandibular curve of Wilson, indicated the necessity for skeletal expansion. At the same time, this method would resolve its crossbite, resulting in improved smile aesthetics. Correction of the crowding, the control of the patient’s vertical growth and coordinating both arches were the secondary outcomes.

### 4.3. Treatment Strategy

To perform the transverse correction without inducing dental compensation on the permanent dentition some clinical choices has been taken.

Performing upper expansion by the traditional method, anchored on the maxillary molars, runs the risk of causing unwanted dental effects such as buccal tipping and extrusion of U6, which would exacerbate the patient’s hyperdivergent characteristics and labial incompetence. To make the proper choice, the patient’s CBCT tac cone-beam was analyzed, to stadiate the midpalate suture [10]. The patient in this case report belongs to stage C, in which the suture appears partially interdigitated; there are two parallel, jagged, high-density lines that are very close together, separated by small spaces, and alternating with an area of low bone density (Figure 7).

An initial diagnosis of stage C indicates an uncertain prognosis for performing a traditional rapid expansion, as the start of interdigitation of the palatine portion of the suture may be imminent [15]. The literature suggests performing palatal expansion by MARPE. This procedure produces less load on the periodontal ligament of the teeth to which it is anchored, because the force expressed by the activation protocol [17], is discharged predominantly to the bone rather than to the dentition, with a significant decrease in overall dental compensations compared with traditional maxillary expansion [15]. 

Second, the crowding and lips incompetence needed to be corrected. The patient was recommended extraction of the upper and lower first premolars, followed by using fixed appliances to achieve efficient space closure. This therapy would shorten treatment time while enhancing the patient’s verticality and correcting crowding at the same time. The patient, however, declined this treatment choice since she wanted non-extraction and less invasive therapy.

Therefore, treatment with MARPE, followed by a fixed appliance, was chosen.

### 4.4. Treatment Progress

Treatment began with the bonding of the mandibular arch, using the In-Ovation system; the goal of this first phase was to remove the dental offsets in the upper arch and level the curve of Wilson so that the true view of the transverse discrepancy present could be obtained (Figure 8).

Mesial and distal stripping was performed on all dental elements in the lower arch. After about 4 months, the correction of the lower arch was almost complete. Therefore, we proceeded with the digital plan in the upper arch.

Following the Easy Driver protocol, the CBCT scan of the maxilla was digitally superimposed with the digital model, to identify the most suitable palatal insertion sites (Figure 9) [17,39].

Two self-drilling BENEfit^®^ miniscrews, 2.0 mm in diameter and 9.0 mm in length, were selected and a surgical CAD-CAM guide was designed and printed to allow guided insertion of the miniscrews (Figure 10). 

After putting the anesthesia paramedian, in the third palatine rugae area, the surgical template was fitted in the maxillary arch, to verify the correct adherence. Subsequently, TADs were easily inserted using a slow-speed contra-angle handpiece. The Hybrid Hyrax was securely attached to the mini-implants with the fixation screws. After two weeks of acclimatization, expansion was begun with five activations in one day, and subsequently one activation per day for 20 days [40,41,42]. At the end of the expansion procedure, the patient was monitored for three months.

Once the control period was ended, a multibraces appliance was inserted in the upper arch and the Hybrid Hyrax was blocked with composite resin inside the expansion screws. The screw was not disconnected from the bands on UL6 and UR6 for most of the treatment; in this way, it was possible to achieve a posterior maximum anchorage to avoid extrusion of the first molars and to finalize properly the case. When most of the corrections were performed, the Hyrax was removed and a transpalatal bar was inserted, to achieve de-rotation of molars most effectively. Class II elastic was used at the end of treatment to improve the anterior projection of the mandible (Figure 11).

After 20 months, the therapeutic goals had been achieved. Facial photographs show improved aesthetics of the smile, with decreased buccal corridors; labial incompetence is mildly improved, as mandibular projection. Even though, circumoral muscle strain on lip closure is not completely resolved, due to the vertical growth of the maxilla that occurred during the treatment, as showed by the augmented gingival smile of the girl.

Intraorally, the following were obtained: a molar and canine Class I relationship, good centering of the dental midlines, normalized overbite, and overjet. The transverse expansion obtained was about 5, 6 mm. The Spee and Wilson curves were leveled, with an absence of the signs of dental compensation present at the start of treatment (Figure 12 and Figure 13).

The maxillary expansion and crowding correction obtained by the end of therapy can be seen by superimposing and comparing pre-and post-treatment digital maxillary models. (Figure 14).

Post-treatment cephalometric values show that the divergence value improved (S-Goc/N-Me, 61.8%), with almost the same proclination of both maxillary (U1-PP, 111.2%) and mandibular (IMPA, 97.8%) incisors (Figure 15; Table 2). On the panoramic radiograph, there are no bone anomalies or indications of apical resorption, confirming proper root parallelism (Figure 15).

Because of the greatest anchoring of U6 and the maintenance of a good incisal tilt, the overall tracing superimposition demonstrates that the patient’s vertical growth was well controlled (Figure 16).

## 5. Discussion

The goal of this paper was to analyze how palatal expansion is performed in adolescent patients, and the effects of various palatal expansion equipment on hard and soft oral tissues.

First described by Angell in 1860 [43], the tooth-borne maxillary expander is still widely used by clinicians to solve transversal discrepancies in the maxilla [23].

Hyrax design appliances are the most commonly used: they present bands on U6 and maxillary first premolars and a 9 mm expansion screw. Mini Hyrax design appliances present only two bands, and the first molars are the anchorage elements; the screw is 8 mm long [26]. Silveira et al. stated that Hyrax and Mini-Hyrax devices are similar in terms of orthodontic and orthopedic effects and patient’s comfort [26].

The force produced when the screw is activated involves the widening of the palatine suture but also has effects on other craniofacial structures, on the upper airways [31,35,36], and soft tissue [24], and produces dentoalveolar changes.

Gökçe [23] compared pre-treatment and post-treatment posteroanterior cephalometric values of adolescents treated with RME and in the Hyrax wearing group observed a mild increase in the maxillary and nasopharyngeal width, leading to an improved breathing function, but also an increase in upper intermolar width. Depending on the stage of maturation of the suture, the expansion obtained in late adolescents is half skeletal and half dental [44] and the force of the apparatus has negative consequences on the anchoring teeth [45,46].

Therefore, expansion with classic tooth-borne devices is frequently related to undesirable implications such as extrusion of the posterior teeth, increase in the buccal angulation of the anchoring elements, dentoalveolar tipping, root resorption of the anchoring elements and contiguous teeth, bone dehiscence, periodontal sequelae such as gingival recessions and loss of alveolar support due to horizontal and vertical reduction in the alveolar ridge [37].

In patients without growth potential, due to the gradual fusion of the median palatine suture, traditional RME expansion (tooth borne) is not very effective [47,48]. Furthermore, the lower the skeletal expansion effect, the more frequent the undesirable effects will be [28].

To perform the expansion and to limit unwanted effects more effectively, there exist alternative devices that use the support of miniscrews inserted in the maxilla to improve the distribution of forces [28,49]. These devices can be bone supported, with only bone anchorage, or both bone and dental support [29].

Mostly of the expander appliances reported in literature are hybrid, characterized by both dental and bone support through miniscrew anchorage [37].

MARPE expansion is a valid alternative to the therapy of transverse discrepancies in the adolescent patient and allows excellent results to be achieved about orthopedic results, with fewer undesirable effects than traditional expansion, and also extends the possibilities of treatment in adult patients [32,49,50].

A systematic review by Kapetanovic et al. demonstrated that MARPE is effective in inducing both skeletal and dental transverse maxillary expansion. However, their study underlined the limited evidence showing that despite its relatively short treatment duration, MARPE may induce dental and periodontal side effects and affect peri-oral soft tissues [16].

Prospective randomized studies by Chun et Al. and Jia et Al., compare conventional RME with MARPE evaluating skeletal, dentoalveolar, and periodontal changes. The rate of success of opening the median palatine suture is higher in assisted expansion with miniscrews (Chun 95%, Jia 100%) compared to that with dental support (Chun 90%, Jia 86.7%) [28,37].

This is because dental anchorage alone may be insufficient to open suture in a patient during the post-pubertal growth phase [28,51]. According to Melsen histological studies, the suture begins to fuse and becomes interdigitated at 16 years for women and 18 years for men [52]. The MARPE treatment showed a more substantial increase in upper jaw width than the RME group [28].

Through multiple skeletal measurements taken with the aid of CBCT, it was seen that after the active expansion phase the skeletal expansion values measured on several landmarks are greater in patients who underwent MARPE [37].

During the consolidation phase, in which the appliance was maintained passive in the mouth to stabilize the suture, the MARPE group showed a minor reduction in transverse widths while the RME group had a major relapse [37]. With both devices, it showed a higher maxillary diameter (MW) at the level of the molar (M-MW) and the premolars level (P-MW), with greater values detected in the group of subjects treated with MARPE [37].

The vestibular tipping value of the U6 results is about half in MARPE patients, compared to patients treated with Hyrax, this implies a lower vestibular displacement of the anchoring elements in hybrid devices concerning tooth-borne devices [28].

The main adverse effect during RME treatment is root resorption. Several studies have focused on the assessment of this parameter and the comparison between resorption in patients treated with MARPE and with traditional RPE.

The quantity of lost on the root surfaces of the anchoring elements is significantly greater in tooth-borne devices than in bone-borne devices, especially on the middle third and the apical third of the root [38,53].

Vestibular bone resorption at the level of the maxillary molars is lower in patients who undergo treatment with hybrid devices. Although bone ridge thickness at the palatal level increases regardless of device type, at the vestibular level vertical and horizontal bone resorption is lower in MARPE patients; this correlates with a lower risk of bone dehiscence [37,50].

Root resorption is an inevitable consequence of dental movement, but tooth-borne devices are responsible for a more consistent reduction in root volume because the resulting expansion forces are directly transmitted to the teeth on which the device is anchored. Furthermore, root resorption was also detected in the elements adjacent to the anchoring teeth, probably due to a discharge of residual stress forces on these elements [36].

The correction of the transverse discrepancy determines changes at the skeletal and dentoalveolar levels, which is also accompanied by an important modification factor for the soft facial tissues [54,55].

Akan et al. stated that the expansion caused an improvement in the values of nose width, mid-facial width, the ratio between both lip length, lower and upper lip angle, lower anterior facial height, and mandibular angle. The modifications are statistically equivalent between RME and MARPE, except for the length of the lower lip which showed a more significant increase with hybrid devices [24].

The expansion also causes changes in the upper airways. Transverse upper jaw deficiency can cause airway stenosis and other complications associated with abnormal tongue posture, and it is a major determinant of obstructive sleep apnea syndrome (OSAS) [20,56,57]. RME can cause an enhancement in the volume and width of the nasopharynx, improving the flow of the upper airways [47]. Considering these results, the rapid expansion of the palate should be considered one of the treatments of choice for patients with oral breathing, and sleep disorders (OSAS).

The transverse modification of the upper arch can also affect the lower arch. The effects are transmitted between the two arches through two main mechanisms: muscular and dental. The expansion of the upper arch changes the balance of the muscular forces of the tongue and cheeks [31,58]. The cheek muscles are moved away from the lower arch, reducing the application of centripetal forces (such as a lip-bumper), favoring the action of the centrifugal force applied by the tongue [47,48,49].

The dental effect occurs in patients who have a posterior crossbite. The expansion movement causes occlusal interference: the vestibular cusps of the upper molars exert a vestibular force on the molar fossa of the lower molars, favoring an enhancement in the width of the mandibular arch [51,59,60,61].

Although the expansion of the upper arch is responsible for a buccal inclination of the lower molars and an increase in the mandibular intermolar width, the values are not clinically significant and are not sufficient to determine a space gain in the perimeter of the mandibular arch [31,62].

According to our study, expansion supported by miniscrews plays a fundamental role in reducing the undesirable effects of RME.

The dental, skeletal, and periodontal effects of surgically assisted expansion cause a lower risk of periodontal damage, while traditional devices (tooth borne) are associated with a greater risk of root resorption, dental tipping, and loss of alveolar support [36].

MARPE is more predictable and determines a more significant expansion of the suture than expansion with Hyrax [28].

As a result, devices that exploit anchoring with miniscrews can represent a valid alternative for the treatment of transverse deficits in patients in the post-pubertal growth phase.

## 6. Conclusions

According to the studies gathered and analyzed in this systematic review, increasing maxillary width with MARPE devices is not only effective, but also correlates with a reduction in side effects associated with standard maxillary expanders.

The force produced by screw activation widens the palatine suture and affects other craniofacial structures, including the upper airway and soft tissue, as well as causing dentoalveolar changes.

Expansion with traditional dental devices is frequently associated with negative consequences such as posterior tooth extrusion, increased vestibular angulation of anchor teeth, dentoalveolar tipping, root resorption of anchor teeth and contiguous teeth, bone dehiscence, periodontal sequelae such as gingival recession, and loss of alveolar support due to horizontal and vertical reduction in the alveolar ridge.

Traditional (tooth-borne) devices are associated with higher risks of side effects such as root resorption, tooth tipping, and loss of alveolar support, whereas surgically assisted expansion has shown dental, skeletal, and periodontal effects with a lower risk of periodontal damage. The amount of loss at the root surfaces of anchorage elements in tooth-borne devices is significantly greater than in bone-borne devices.

Furthermore, the vestibular tipping value of the U6 results is approximately half that of Hyrax-treated patients, implying that hybrid devices have less vestibular displacement of anchor elements than tooth-loaded devices.

In conclusion, MARPE is more predictable than Hyrax expansion and results in greater suture expansion with fewer adverse effects. Miniature anchorage devices may be a viable option to address transverse deficits in post-puberty patients.

Surgically assisted expansion is a thriving field of research with widespread interest in the scientific literature. Although the findings of published studies on this topic so far indicate a promising use of the new expansion devices, more in-depth, high-quality studies in the form of randomized clinical trials and prospective cohort studies with a well-defined device design and treatment protocol are required to provide a higher quality of evidence on the efficacy of surgically assisted expansion.

## Figures and Tables

**Figure 1 children-09-01046-f001:**
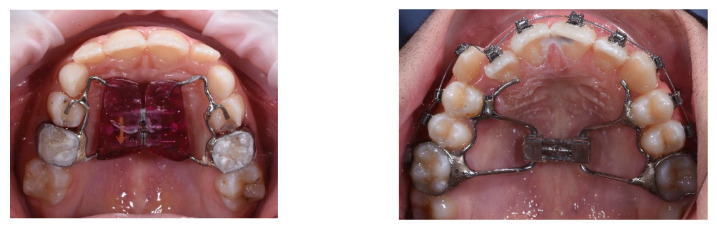
Conventional appliance RME: TTB RME and TB RME.

**Figure 2 children-09-01046-f002:**
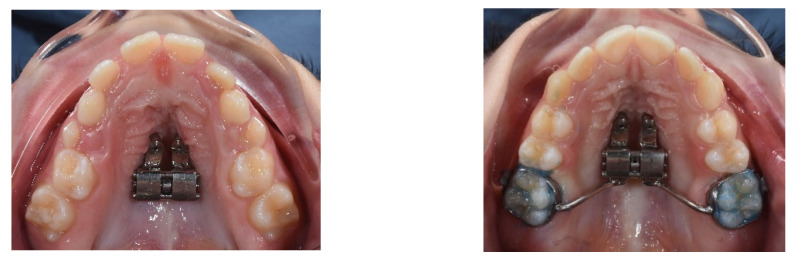
MARPE appliance: BB MARPE and TBB MARPE.

**Figure 3 children-09-01046-f003:**
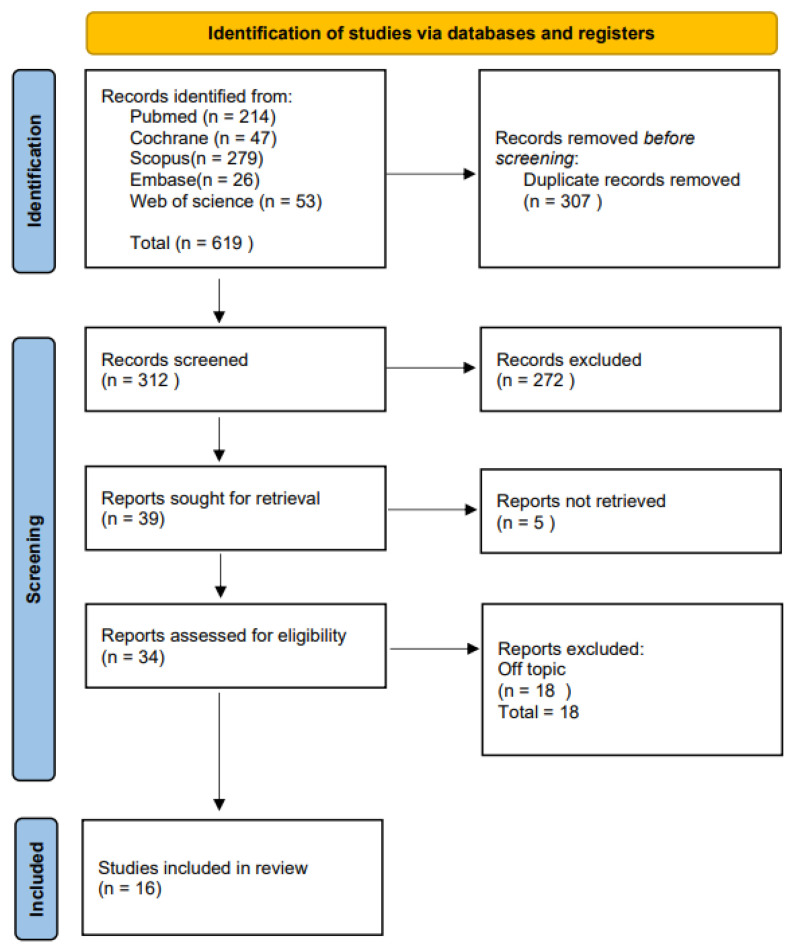
Literature search Preferred Reporting Items for Systematic Reviews and Meta-Analyses (PRISMA) flow diagram.

**Figure 4 children-09-01046-f004:**
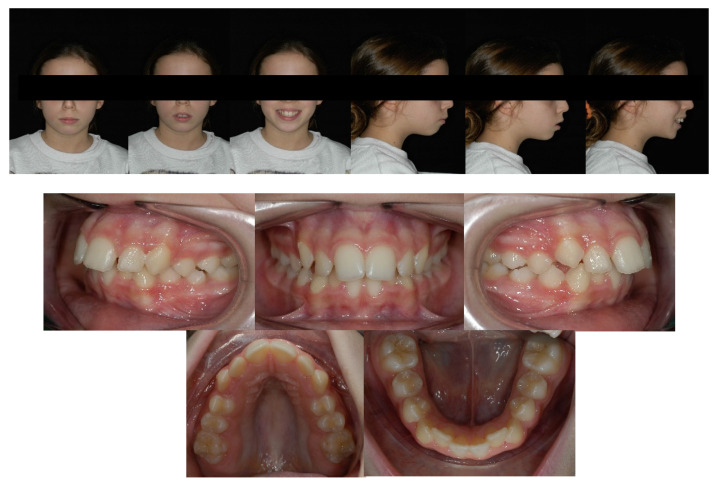
Pre-treatment photographs.

**Figure 5 children-09-01046-f005:**

Pre-treatment digital models registered the centric dental position.

**Figure 6 children-09-01046-f006:**
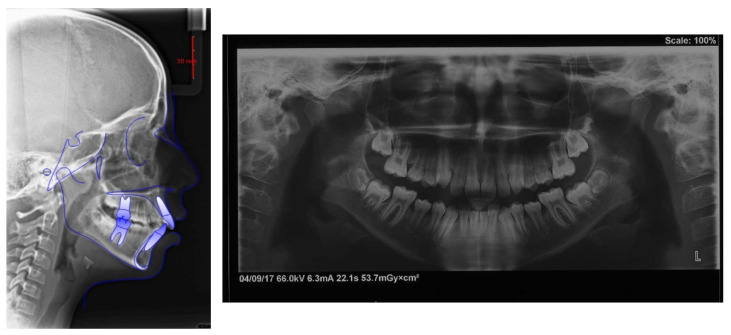
Pre-treatment lateral cephalogram and panoramic radiograph.

**Figure 7 children-09-01046-f007:**
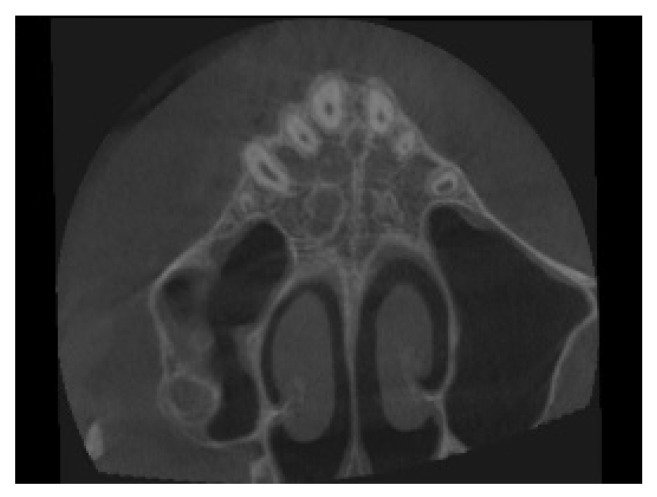
Through CBCT the morphology of the midpalatal suture, can be evaluated. In this case, identified as stage C.

**Figure 8 children-09-01046-f008:**
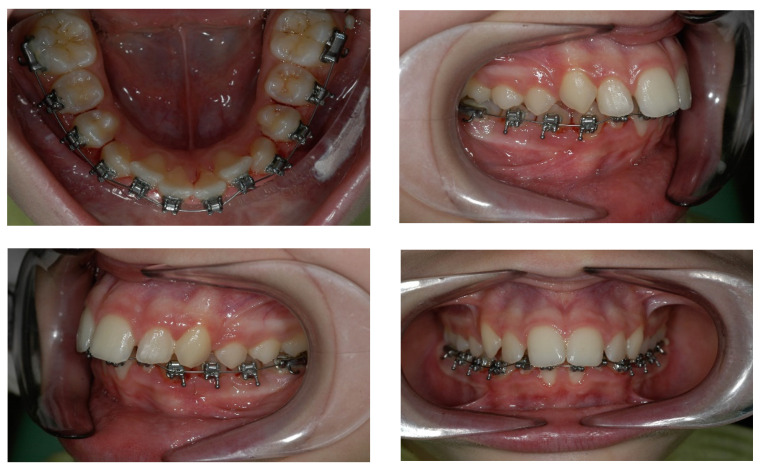
Bonding of the lower arch using the In-Ovation system.

**Figure 9 children-09-01046-f009:**
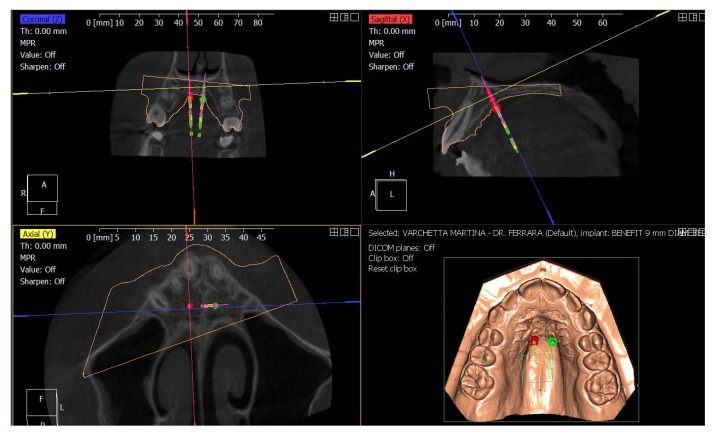
Digital planning with Easy Driver^®^ protocol.

**Figure 10 children-09-01046-f010:**
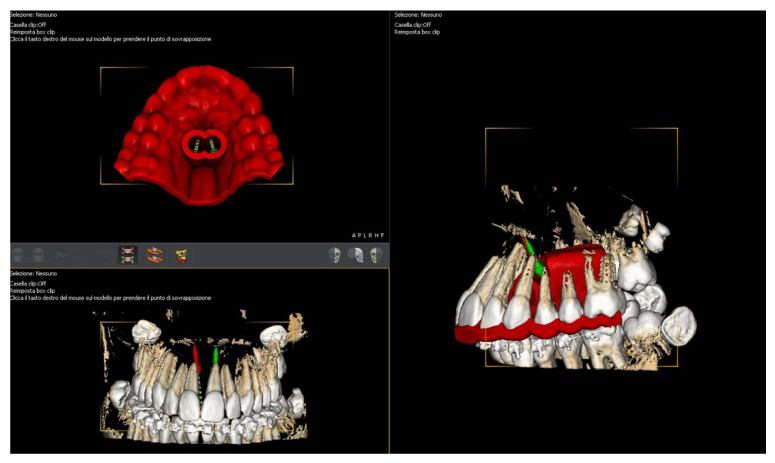
Digital plan of the TADs insertion guide.

**Figure 11 children-09-01046-f011:**
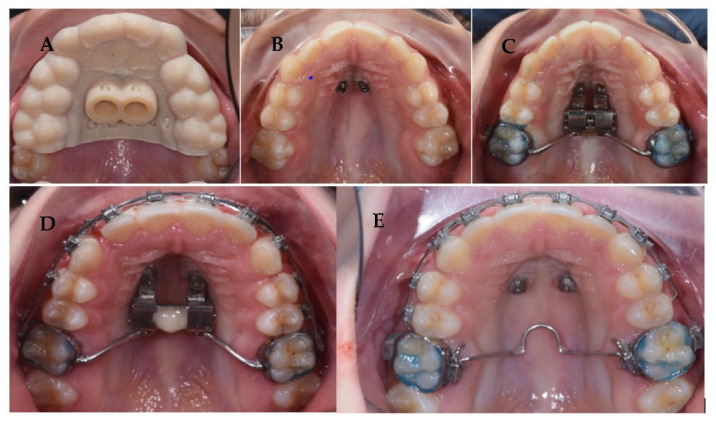
Treatment flow:(**A**) Miniscrew guide fitting; (**B**) Occlusal view of miniscrew; (**C**) Appliance positioning; (**D**) Occlusal view after active expansion; (**E**) The finishing stage of treatment.

**Figure 12 children-09-01046-f012:**
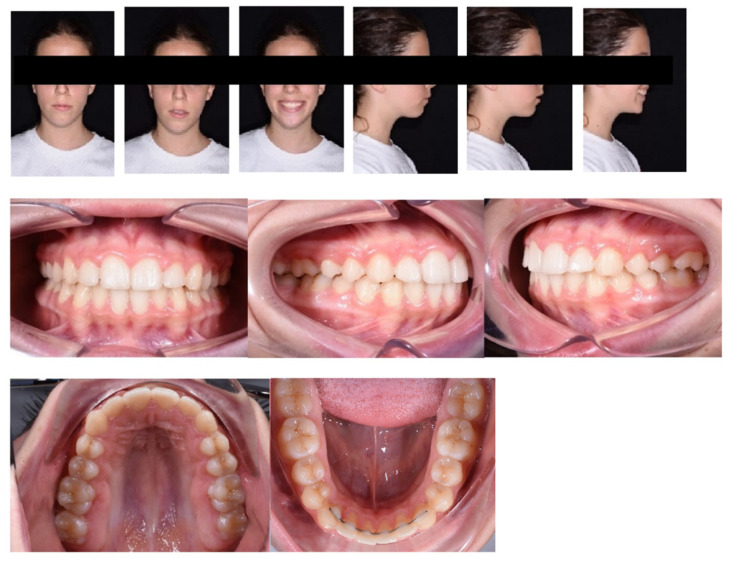
Post-treatment facial and intraoral photographs.

**Figure 13 children-09-01046-f013:**
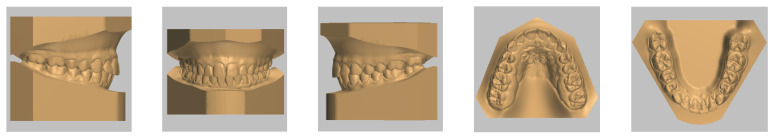
Post-treatment digital model.

**Figure 14 children-09-01046-f014:**
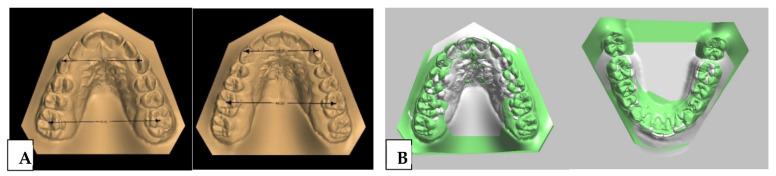
(**A**) Comparison and (**B**) superimposition of digital maxillary models before and after treatment.

**Figure 15 children-09-01046-f015:**
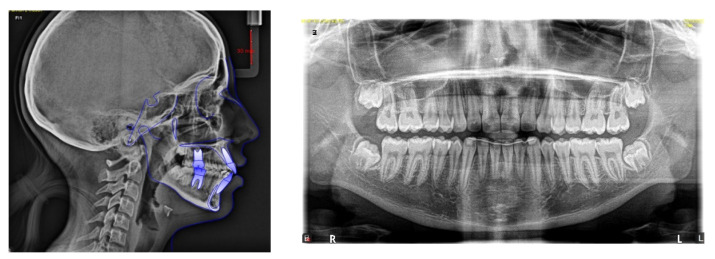
Post-treatment lateral cephalogram and panoramic radiograph.

**Figure 16 children-09-01046-f016:**
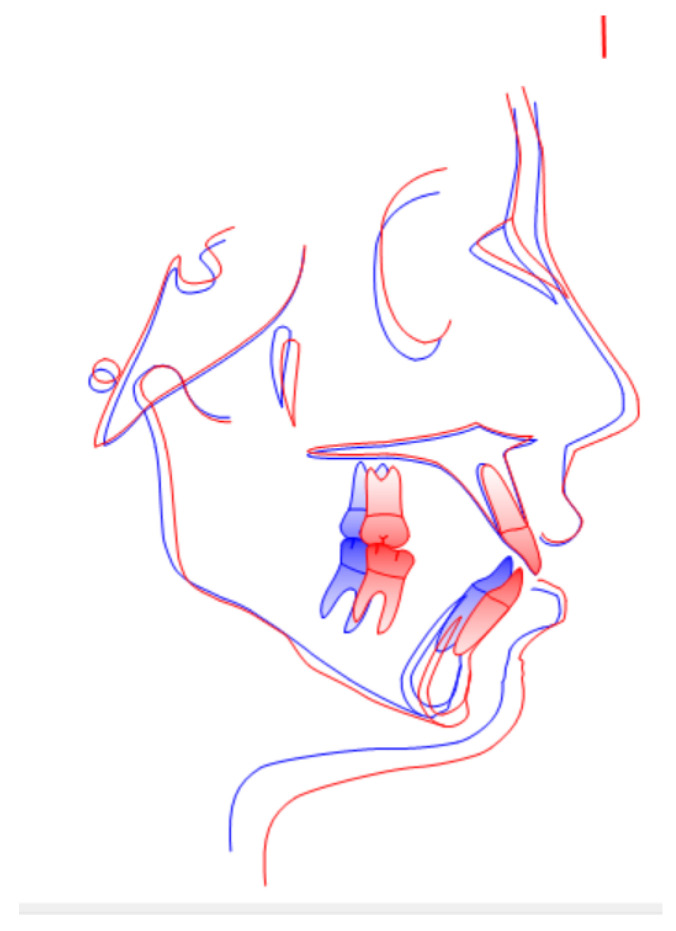
Cephalometric superimpositions (blue, pre treatment; red, post treatment).

**Table 1 children-09-01046-t001:** Summary of selected articles data.

Author/ Year	Study Design	Sample Size	Data Collection	Average Age (Years Old)	Type of Appliance	Protocol of Expansion	Features of the Screw	Amout of Expansion Achieved (mm)	Outcomes	Retention Period
Gökçe, 2021	Retrospective	54 (25 M, 29 F)	Pretreatment (T0) and posttreatment (T1) Postero Anterior cephalometric radiographs	13.28 ± 1.20 (TBB) 13.08 ± 1.06 (TTB) 12.05 ± 1.35 (TB)	-TBB -TTB -TB	First week 2 turns/day and then 1 turn/day 1 turn = 0.25 mm	9 mm Hyrax expansion screw (G&H Orthodontics, Franklin, IN, USA)	Intermolar width: TB = 5.5 mm TTB = 4.47 mm BB = 6.71 mm	Skeletal changes are more evident in TTB and TBB groups	3 months
Yildirim, 2018	Observational	20 (11 F, 9 M)	Micro CT device on premolar teeth after expansion and extraction	11–16 (mean age 14.31 ± 1.36)	BB and TTB in the same patient using modified device	N.D.	Hyrax screw	N.D.	Root resorptions are more frequent in the TTB group, mostly in the apical and middle thirds	3 months
Canan, 2017	RCT	47 (25 F, 22 M)	Superimposition of 3D digital maxillary dental models; OHIP-14 questionnaire to value quality of life	12.63 ± 1.36 (TB) 12.92 ± 1.07 (BB) 13.41 ± 0.88 (HB)	-TB -BB -TBB	2 turns/day 1 turn = 0.25 mm	9 mm Hyrax; Lewa-Dental, Remchingen, Germany)	Mean turns = 26; Mean screw expansion = 6.5 mm	Dentoalveolar maxillary expansion with mild relapse in all groups; Lower expansion of the BB on the right side; Spontaneous interdental expansion in the mandibular dentitions	6 months
Silveira, 2021	RCT	34	Digitally superimposed pre-treatment and post-retention 3D intraoral scans on the palatal rugae using the software 3DSlicer	11–16 year	-Hyrax (TB) -Mini-hyrax (TB)	2 turns/day 1 turn = 0.25 mm	−8 mm mini expander jackscrew (Dynaflex, Saint Ann, USA) −9 mm Hyrax jackscrew (Morelli, Sorocaba, Brazil)	Mean turns = 30; Mean screw expansion = 7.5 mm	No significant differences in dental effects, impact on quality of life and pain perception	6 months
Kavand, 2019	Retrospective	36	CBCT at T0 before expansion and T1 post retention	14.7 years (BB) 14.4 years (TB)	-BB -TB	2 turns/day 1 turn = 0.25 mm	Jackscrew (Palex II Extra-Mini Expander, Su mmit Orthodontic Services, Munroe Falls, OH, USA)	Mean palatal width expansion TB = 1.5 ± 0.4 BB = 2.2 ± 0.3 mm	Increased volume of nasal cavity and nasopharynx; Increased maxillary dental and skeletal width in both groups; Buccal tipping of maxillary molars in TB.	3 months
Alcin, 2021	RCT	20 (12 F, 8 M)	Micro-CT of maxillary first premolars	12–15	-TBB -TB -Acrylic-bonded TB -Full-coverage TB	1turn/day 1 turn = 0.25 mm	Hyrax screw	Mean turns = 34;	All expansion appliances cause root resorption, mostly on the buccal surface; TBB causes lesser root resorption than TB appliances.	3 months
Celenk-Koca, 2018	RCT	40 TB group: 12 F, 8 M; BB group 13 F, 7 M	CBCT evaluation of: -transverse skeletal widths; -buccal bone thickness; -tooth inclination -root length	13.84 ± 1.36 (TB) 13.81 ± 1.23 (BB)	-TB -BB	2 turns/day 1 turn = 0.25 mm	Hyrax screw	Molar width: TB = 4.2 + 1.7 BB = 4.5 + 1.3	BB increased the maxillary suture opening more than 2.5 times than TB and did not result in any dental side effects	6 months
Annarumma, 2021	Retrospective	24 (12 M, 12 F)	CBCT evaluation of: -maxillary width; -inclination of the alveolar process; -tooth inclination; -vertical dental height; -periodontal tissues	13.9	-BB on 4 miniscrews	2 turns/day	Hyrax screw on BB	mean expansion screw = 8.12 ± 2.98 mm	BB expansion was effective with negligible dental effects	12 months
Aljawad, 2021	Case-control retrospective study	33 17 TB 16 control group	Upper airway dimensions (CBCT)	mean age 12.6 ± 1.8	-TB	2 turns/day 1 turn = 0.25 mm	Hyrax screw (Dentaurum, Ispringen, Germany)	N.D.	RME causes an increase in upper airway dimensions	3 months
Torun, 2017	Retrospective study	28 (10 M, 18 F)	Measurements of soft tissue with CBCT at T0, T1	13.91 ± 1.8	TB	2 turns/day 1 turn = 0.25 mm	Hyrax screw (Dentaurum, Ispringen, Germany)	Mean Screw expansion: 9–10 mm	significant changes in facial soft tissues P < 0.001	6 months
Lotfi, 2018	Retrospective study	20 (8 M, 12 F)	Measurements of upper airway volume changes with CBCT at T0, T1	12. 3 ± 1.9	-TB	2 turns/day	Hyrax screw	N.D	Significant changes in nasal cavity volume	6 months
Chun, 2022	Prospective RCT	40 TB group: 20 (6 M, 14 F) MARPE group (8 M, 12 F)	CBCT evaluation of: -skeletal changes; -dentoalveolar changes; -periodontal changes	14.0 ± 4.3 years	-TB -MARPE	1 turn/day for 35 days 1 turn = 0.20 mm	Hyrax expander (Dentaurum, Ispringen, Germany)	7 mm screw expansion	Greater increase in nasal width in the molar region (M-NW) and greater palatine foramen (GPF) in the MARPE group compared to the TB group; Similar dentoalveolar changes except for the maxillary width (PM-MW, M-MW). The MARPE group presented greater bilateral first premolar (PM-MW) and molar (M-MW) maxillary width in relation to the TB group; Lesser buccal displacement of the anchor teeth in the MARPE group.	3 months
Jia, 2021	Prospective RCT	60 MARPE 30 TB 30	CBCT and dental cast	14.8 ± 1.5 (TB) 15.1 ± 1.6 (MARPE)	-MARPE -TB	2 turns/day 1 turn = 0.25	MARPE TTB: jackscrew (length: 12 mm; anatomic expander type: ‘‘s;’’ Forestadent, Pforzheim, Germany) TB HYRAX: jackscrew (anatomic expander type; Forestadent)	Mean expansion at maxillary basal bone TTB: 4.53 TB: 4.53	MARPE enabled more predictable and greater skeletal expansion, less buccal tipping and alveolar height loss on anchorage teeth.	3 months
Akan, 2021	RCT	32 16 TB 16 TBB	Changes in soft tissues before RME (T0) and post-retention (T1) evaluated by stereophotogra mmetry	13.4 ± 1.3	-TB -TBB	First week 2 turns/day and then 1 turn /day 1 turn = 0.25 mm	TB:Hyrax screw (Dentaurum, Ispringen, Germany) MARPE TTB mini-screws 2 mm diameter and 9 mm length (Benefit mini-implants; PSM Medical Solutions)	Mean numbers of activations were 25.25 ± 4.42 turns in hyrax group and 24.88 ± 3.40 turns in hybrid hyrax group.	Both appliances had effects on soft tissue profile; Anterior face height and lower face height increased in both groups; Upper lip length increased by 0.36 mm in theTBB group and 0.10 mm in the TB group.	3 months
Lo Giudice, 2020	Retrospective study	33	Linear and angular measurements in the coronal view to assess buccal inclinations and widths of mandibular posterior units.	14.4 ± 1.3 (TB) 14.7 ± 1.4 (BB)	-TB -BB	2 turns/day 1 turn = 0.25 mm	Hyrax miniscrew (Palex II ExtraMini Expander, Su mmit Orthodontic Services, Munroe Falls, OH, USA; Figure 1B)	16–26 width TB = 4.20 ± 1.39 BB = 3.02 ± 1.48	A clinically significant gain of space in the mandibular arch should not be expected after RME	6 months
Cheung, 2021	RCT	44	Measurements of upper airway volume changes with CBCT at T0, T1	ND	-TB -TBB	2 turns/day 1 turn = 0.25 mm	Hyrax screw (Hyrax, Dentaurum, Ispringen, Germany) Keles keyless expander (Keles, Istanbul, Turkey)	N.D.	No statistically significant difference across the TB and TBB	6 months

**Table 2 children-09-01046-t002:** Cephalometric measurements pre and post treatment. A: Point A; B: Point B; S: Saddle; N: Nasion; Pg: Pogonion, SN: Sella-Nasion Plane, MP: Mandibular plane; U1: Upper central incisor; IMPA: incisor mandibular plane angle.

Measurements	Norm	Pre-Treatment	Post-Treatment
**Anterior Cranial Base(S-N)**	74.5 ± 3	66	66.5
**Facial Axis (BaN-PTGn)**	90° ± 3	84.4°	85.8°
**Mandibular Lenght (Goc-Me)**	74.5 ± 5	61.9	64
**Posterior Cranial Base (S-Ar)**	34 ± 3	30.9	31.4
**Height Mandibular Ramus (Ar-Goc)**	47.5 ± 5	33.9	38.3
**Saddle angle: (N-S-Ar)**	123° ± 5	120.4°	118.8°
**Articular angle (S-Ar-Goc)**	143° ± 7	150.1°	149.6°
**Gonial angle (Ar-Goc-Me)**	130° ± 7	128.6°	129.1°
**Upper gonial angle (Ar-Goc-N)**	52° ± 3	53°	52.4°
**Lower gonial angle (Me-Goc-N)**	70° ± 2	75.6°	76.7°
**Anterior facial height (N-Me)**	113 ± 7	105.2	108.9
**Posterior facial height (S-Goc)**	77.5 ± 7.5	62.6	67.3
**Jarabak Facial Proportion %**	61% ± 3	59.5%	61.8%
**SNA**	82° ± 2	80.9°	79.7°
**SNB**	80° ± 2	75.1°	76°
**ANB**	2° ± 2	5.8°	3.7°
**U1—palatal plane**	110° ± 2	112.7°	111.2°
**IMPA**	90° ± 3	96.9°	97.8°

## Data Availability

All experimental data to support the findings of this study are available by contacting the corresponding author upon request.

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
