# Peer review of "Rapid Maxillary Expansion on the Adolescent Patient: Systematic Review and Case Report"

_children, 2022, doi:10.3390/children9071046_

Round 1

Reviewer 1 Report

Paper is not well organised. And does not follow a standard structure (introduction, material and methods, results, discussion and conclusions). However, some parts of presented paper are not understandable or require edition and better description. When the paper  has 26 authors (!), the quality of text should be almost perfect.

Please see remarks described below.

Major remarks:

The introduction section should be developed – I suggest to include the description of changes in midpalate suture after RPE on the basis of literature.

My greatest hestitations are connected with literature analysis, Some papers do nor include comparable data according to treatment methods. For instance – the protocol of activation .. There’s no information about achieved overcorrection, occurred relapse and retention phase. What kind of screw was used – it was a typical hyrax screw? In which size? Please provide the manufacturer’s data including milimeters of expansion per one activation.

The data presentation in two huge tables is hard to read – please improve the visibility of obtained data

The CBCT figures showing the relevant points used to perform measurements  significantly increase the paper’s receival.

The number of references should be extended, the reference list requires editing, according to ADA citation style (for example capital letters in journals’ titles)

Conclusions section is poor in content, there is a lack of clinical signification of obtained results and perspectives for future research

Furthermore, please consider the discussion section – should it be after case report, at the final part of the paper, just before conclusions, shouldn’t it?

There are other, mostly minor remarks:

Line 50– unnecessary capital letter

Line 63– unnecessary capital letter

Line 64 – too long sentence, hard to understand

Line 70– midpalatal suture

Line 71 – did you mean chronological age?

Line 72 Beam (should be written with capital letter)

Line 75 “which is not good connector – the sentence sound illogical

Line 76 – the difference exists, but it is not a rule – so instead of “have” should be “can/could have”

Line 77 – ‘Youth” – what did you mean?

Line 80 – require further explanation

Line 88 – is the “resolution” the appropriate word?

Line 90 – in comparison to what?

Line 109 – the sentence should be corrected, i.e the inclusion criteria were…..

Line 114 – is the’ satisfy a proper word (better i.e fulfill)

Line 140 what does it mean “outcome of interest”

Line 135 – should be figure 3

Line 154 – the activation protocol depend also on the type of the screw – please provide such information

Line 171 should be most commonly used…

Line 175 – the sentence is hard to understand

Line 181 instead of “higher” should be i.e an increase in….

Line 202 the sentence is hard to understand

Line 208 -lack of reference

Line 220 –“expansion phase” - did you mean an active phase?

Line 321  -the aim of expansion is first of all to correct the tranversal discrepancy. The crowding release is a secondary effect not the main goal

Line 345 the sentence is hard to understand

Line 347 – what does “definitive’ mean?

Line  348  - instead of’ vestubulo version”should be i.e buccal tipping,or vestibular inclination

Author Response

Paper is not well organised. And does not follow a standard structure (introduction, material and methods, results, discussion and conclusions). However, some parts of presented paper are not understandable or require edition and better description. When the paper has 26 authors (!), the quality of text should be almost perfect.

Please see remarks described below.

Major remarks:

The introduction section should be developed – I suggest to include the description of changes in midpalate suture after RPE on the basis of literature. –

It has been added.

My greatest hestitations are connected with literature analysis, Some papers do nor include comparable data according to treatment methods. For instance – the protocol of activation .. There’s no information about achieved overcorrection, occurred relapse and retention phase. What kind of screw was used – it was a typical hyrax screw? In which size? Please provide the manufacturer’s data including milimeters of expansion per one activation.

Thanks for your suggestions. Table 1 has been implemented with more detailed informations.

 The data presentation in two huge tables is hard to read – please improve the visibility of obtained data

It has been modified.

The CBCT figures showing the relevant points used to perform measurements significantly increase the paper’s receival.

Thank you.

The number of references should be extended, the reference list requires editing, according to ADA citation style (for example capital letters in journals’ titles)

It has been modified.

Conclusions section is poor in content, there is a lack of clinical signification of obtained results and perspectives for future research

It has been corrected.

Furthermore, please consider the discussion section – should it be after case report, at the final part of the paper, just before conclusions, shouldn’t it?

It has been corrected.

There are other, mostly minor remarks:

Line 50– unnecessary capital letter

It has been corrected.

Line 63– unnecessary capital letter

It has been corrected.

Line 64 – too long sentence, hard to understand

It has been corrected.

Line 70– midpalatal suture

It has been corrected.

Line 71 – did you mean chronological age?

Yes, we mean the chronological age of patient.

Line 72 Beam (should be written with capital letter)

It has been corrected.

Line 75 “which is not good connector – the sentence sound illogical

It has been corrected.

Line 76 – the difference exists, but it is not a rule – so instead of “have” should be “can/could have”

It has been corrected.

Line 77 – ‘Youth” – what did you mean?

 We mean that even if the patient is very young, the degree of ossification of the midpalatal suture is more advanced than expected. It has been rephrased.

Line 80 – require further explanation

It has been improved.

Line 88 – is the “resolution” the appropriate word?

It has been corrected.

Line 90 – in comparison to what?

In comparison to perform miniscrew insertion without digital planned solution.

Line 109 – the sentence should be corrected, i.e the inclusion criteria were…..

The sentence has been rephrased.

Line 114 – is the’ satisfy a proper word (better i.e fulfill)

It has been corrected.

Line 140 what does it mean “outcome of interest”

The papers did not meet the inclusion criteria. It has been corrected.

Line 135 – should be figure 3

It has been corrected.

Line 154 – the activation protocol depend also on the type of the screw – please provide such information

It has been corrected.

Line 171 should be most commonly used…

It has been corrected.

Line 175 – the sentence is hard to understand

It has been corrected.

Line 181 instead of “higher” should be i.e an increase in….

It has been corrected.

Line 202 the sentence is hard to understand

It has been rephrased.

Line 208 -lack of reference

It has been corrected.

Line 220 –“expansion phase” - did you mean an active phase?

Yes. It has been corrected.

Line 321  -the aim of expansion is first of all to correct the tranversal discrepancy. The crowding release is a secondary effect not the main goal

It has been rephrased.

Line 345 the sentence is hard to understand

It has been rephrased.

Line 347 – what does “definitive’ mean?

It has been corrected.

Line  348  - instead of’ vestubulo version”should be i.e buccal tipping,or vestibular inclination

It has been corrected.

Reviewer 2 Report

This paper evaluated “The maxillary expansion on the adolescent patient: systematic review and case report.

Generally, this manuscript is interesting. However, there are some concerns as presented and some of these are discussed below.

Comments:

1.     The manuscript was written well.

2.     Why are there so many authors when this paper is a review and case report?In Discussion section of the review, authors should Put together paragraphs.

3.     In case report, the patient is still growing at the age of 11, are you considering it?

4.     I think the patient is 11 years old and the bones are not mature, is there any problem with the stability of the anchor?

5.     Is there a CBCT image after expansion?

6.     There is a possibility of relapse, but what do you think? What kind of measures are you thinking about?

Author Response

Generally, this manuscript is interesting. However, there are some concerns as presented and some of these are discussed below.

Comments:

 The manuscript was written well.

Thank you

  1. Why are there so many authors when this paper is a review and case report? In Discussion section of the review, authors should Put together paragraphs.

All the authors contributed to the realization of the work. The contributions of each author are given at the end of the article.

The discussion has been improved as requested.

  1. In case report, the patient is still growing at the age of 11, are you considering it?

Yes, but the patient showed permanent dentition, and we wanted to perform the correction of the transversal discrepancy without inducing dental compensation. For this reason we perform the treatment plan described.

  1. I think the patient is 11 years old and the bones are not mature, is there any problem with the stability of the anchor?

No, the miniscrew were left in place for all the treatment period (20 months). We didn’t  find any anchoring problems.

  1. Is there a CBCT image after expansion?

No, unfortunately we are not in possession of it.

  1. There is a possibility of relapse, but what do you think? What kind of measures are you thinking about?

Yes, the possibility of relapse always exits. The patient is followed up every 6 months and she wear a Schwarz plate every night in order to preserve the treatment outcomes.

Reviewer 3 Report

Dear

the running article needs to be reviewed since some papers are not included.

A recent review of the literature by A. Kapetanovic was not taken into consideration.

Kind regards

Author Response

Dear

the running article needs to be reviewed since some papers are not included.

A recent review of the literature by A. Kapetanovic was not taken into consideration.

Kind regards

Thanks for the suggestion. The paper has been added and references have been improved.

Round 2

Reviewer 1 Report

Authors applied suggested changes. The paper is improved. 

Author Response

Thanks to the reviewer for comments and suggestions.

Reviewer 2 Report

I am now satisfied that all necessary changes have been made.
Further, the comments of other referees were cleared in this revised manuscript.

I recommend acceptance and publication of your research in the Children.

Author Response

I am now satisfied that all necessary changes havebeen made.
Further, the comments of other referees were cleared in this revised manuscript.

I recommend acceptance and publication of your research in the Children.

We are grateful for your suggestions.

Reviewer 3 Report

some additional comments :

Bias because not a single type of MARPE ?

Which criterion was used for evaluating adverse effects? Measurement of relaps? unspecified…

Bias because this article includes non-randomized studies and retrospective studies :

Unjustified exclusion criteria

Author Response

Bias because not a single type of MARPE ?

Which criterion was used for evaluating adverse effects? Measurement of relaps? unspecified…

Bias because this article includes non-randomized studies and retrospective studies :

Unjustified exclusion criteria

Thank you for your comment. Our study, being a systematic review on how expansion is performed in adolescents, we described the various types of devices that allow it, so not only MARPE.

The study, being focused not on adverse effects but on the quantity and quality of expansion, reported side effects found by the studies included in PRISMA only where they were reported by the authors (see discussion and summary table). Thanks to your suggestion more studies on adverse effects and incidence of relaps may follow.

In addition, being a topic of recent interest in the scientific community, and having a limited number of publications, it was necessary to include different types of studies to provide a more comprehensive view of this clinical approach.